# Evaluation and identification of wild lentil accessions for enhancing genetic gains of cultivated varieties

**Mohar Singh**[1]*, **Sandeep Kumar**[2], **Ashwani Kumar Basandrai**[3], **Daisy Basandrai**[3], **Nikhil Malhotra**[1], **Deep Rattan Saxena**[4], **Dorin Gupta**[5], **Ashutosh Sarker**[6], **Kuldeep Singh**[2]

**1** Regional Station, National Bureau of Plant Genetic Resources, Shimla, India, **2** National Bureau of Plant Genetic Resources, Pusa, New Delhi, India, **3** CSK Himachal Pradesh Agriculture University, Palampur, India, **4** Rafi Ahmad Kidwai College of Agriculture, Sehore, India, **5** Faculty of Veterinary and Agricultural Sciences, The University of Melbourne, Melbourne, Australia, **6** South Asia and China Regional Programme, International Centre for Agricultural Research in Dry Areas, DPS Marg, Pusa, New Delhi, India

* singhmohar_2003@yahoo.com

**Data Availability Statement:** All relevant data are within the paper and its Supporting Information files

## Abstract

Domesticated lentil has a relatively narrow genetic base globally and most released varieties are susceptible to severe biotic and abiotic stresses. The crop wild relatives could provide new traits of interest for tailoring novel germplasm and cultivated lentil improvement. The primary objective of this study was to evaluate wild lentil accessions for identification of economically viable agro-morphological traits and resistance against major biotic stresses. The study has revealed substantial variations in seed yield and its important component characters. Further, the diversity analysis of wild accessions showed two major clusters which were bifurcated into sub-clusters, thereby suggesting their wider genetic divergence. However, principal component analysis exhibited that seed yield plant$^{-1}$, number of seeds plant$^{-1}$, number of pods plant$^{-1}$, harvest index and biological yield plant$^{-1}$ contributed significantly to the total genetic variation assessed in wild lentil taxa. Moreover, some of the wild accessions collected from Syria and Turkey regions showed resistance against more than one disease indicating rich diversity of lentil genetic resources. The identification of most promising genotypes carrying resistance against major biotic stresses could be utilized in the cultivated or susceptible varieties of lentil for enhancing genetic gains. The study has also identified some trait specific accessions, which could also be taken into the consideration while planning distant hybridization in lentil.

## Introduction

Lentil is a self-pollinating true diploid (2n = 2x = 14) grain legume crop having genome size of 4063 Mb [1]. The cultivated lentil species (*Lens culinaris*) encompasses two groups, established on the basis of distinct morphological characters, the small-seeded (*microsperma*) and large-seeded (*macrosperma*) [2]. The production and productivity of lentil has increased from an average yield of 565 kg ha$^{-1}$ in 1961–63 to 1153 kg ha$^{-1}$ during 2016–2017 [3]. Despite the

**Funding:** Authors thank the Regional Coordinator of South Asia and China Regional Programme for grant in aid to accomplish the proposed research.

**Competing interests:** Authors declare that they have no competing interests.

**Abbreviations:** NBPGR, National Bureau of Plant Genetic Resources; NARS, National Agricultural Research System; ICARDA, International Centre for Agricultural Research in Dry Areas; PM, Powdery Mildew; FW, Fusarium Wilt; ILWL, International Legume Wild Lentil; PCA, Principle Component Analysis; PC, Principle Component; BIGM, Biodiversity and Integrated Gene Management; CWRs, Crop Wild Relatives; RCBD, Randomized Complete Block Design; UPGMA, Unpaired Group Method Analysis; SMTA, Standard Material Transfer Agreement.

tremendous improvement, the current lentil yield is much below as compared to other pulses. In India, most of the lentil varieties which were developed through pure line selection showed low yield potential. In cultivated lentil varieties, the yield limiting factors are lack of seedling vigour, very high rate flower drop and low pod setting, lack of lodging resistance and exposure to major biotic and abiotic stresses [4]. The pedigree analysis of some important released varieties exhibited that a few donors have contributed substantially into the background of those varieties [5]. Therefore, to attain further breakthrough for enhancing genetic gains, new target traits are needed to be identified and introgressed into cultivated gene pool for widening the genetic base of cultigens. Crop wild relatives (CWRs) are an invaluable reservoir of productivity enhancement related characters having resilience to climate change and farming system, and are source of novel traits [6–7]. The global wild lentil germplasm introduced from the International Centre for Agricultural Research in Dry Areas (ICARDA), has been multiplied, characterized and evaluated against target characters [8]. The promising 96 wild lentil accessions selected from 405 global wild collections were validated under multilocational evaluation for identifying stable donors against the target traits. The present study was therefore, undertaken (1) to evaluate promising wild lentil accessions for various target agronomical characters and major biotic stresses and (2) to identify the potential accessions (gene sources) carrying important traits for enhancing genetic gains of cultivated gene pool.

## Materials and methods

### Plant materials

The experimental genetic material comprised of 96 wild lentil accessions (Table 1) selected from 405 global wild collections including 2 cultivated varieties [8] consisting of *L. orientalis* (24), *L. odemensis* (16), *L. tomentosus* (8), *L. nigricans* (17), *L. ervoides* (24), *L. lamottei* (5) and *L. culinaris* (cultivated) (2) along with two check varieties (Precoz [Argentina], and L830 [India]). These were evaluated under multi-location and multi-season performance for target characters viz. earliness, pod number, seed yield and resistance against rust, powdery mildew and Fusarium wilt under field and controlled screening conditions. The original identity of these accessions was same as mentioned by the Biodiversity and Integrated Gene Management Unit (BIGM) at ICARDA.

### Agronomic evaluation

All 96 wild lentil accessions were evaluated under two agro-ecological locations viz. Experimental Farm of NBPGR Regional Station, Shimla ($28^0$ 35′ N, $70^0$ 18′ E, altitude 2276 m amsl) during winter season of 2014–2015, 2015–2016 and 2016–2017; and CSKHPKV Mountain Agricultural Research and Extension Centre, Sangla ($31^0$ 55′ N, $77^0$ 50′ E altitude 3450 m amsl) during summer season of 2015, 2016 and 2017. Each entry was replicated thrice in 3 m long row and 35 cm apart. The observations were recorded against important agro-morphological characters (days to flowering, days to maturity, plant height, number of branches plant$^{-1}$, number of pods plant$^{-1}$, number of seeds plant$^{-1}$, number of seeds pod$^{-1}$, 100-seed weight, seed yield plant$^{-1}$, and biological yield plant$^{-1}$) in both locations. Further, numerical data was subjected to statistical analysis using SAS software [9]. The coefficient of variation for various characters was determined using the formulae of Burton [10] and broad sense heritability was also calculated as per the method suggested by Lush [11]. The expected genetic advance was performed as per the procedure of Johnson et al. [12].

**Table 1. List of taxa wise wild lentil accessions along with their country of origin.**

| S. No | Taxa/accession | Origin | S. No. | Taxa/accession | Origin |
|---|---|---|---|---|---|
| | *L. orientalis* | | | *L. lamottei* | |
| 1 | ILWL 7 (EC718234) | Turkey | 50 | ILWL 14 (EC718236) | France |
| 2 | ILWL 8 (EC718235) | Turkey | 51 | ILWL 15 (EC718238) | France |
| 3 | ILWL 75 (EC718456) | Israel | 52 | ILWL 29 (EC718251) | Spain |
| 4 | ILWL 89 (EC718470) | Turkey | 53 | ILWL 429 (EC718686) | Spain |
| 5 | ILWL 95 (EC718475) | Turkey | 54 | EC718692 | France |
| 6 | ILWL 96 (EC718476) | Turkey | | *L. odemensis* | |
| 7 | ILWL 101 (EC718479) | Turkey | 55 | ILWL 20 (EC718243) | Palestine |
| 8 | ILWL 117 (EC718488) | Syria | 56 | ILWL 35 (EC718276) | Turkey |
| 9 | ILWL 124 (EC718490) | Syria | 57 | ILWL 39 (EC718277) | Palestine |
| 10 | ILWL 181 (EC718491) | Syria | 58 | ILWL 165 (EC718281) | Syria |
| 11 | ILWL 227 (EC718513) | Syria | 59 | ILWL 166 (EC718282) | Syria |
| 12 | ILWL 230 (EC718515) | Syria | 60 | ILWL 167 (EC718283) | Syria |
| 13 | ILWL 243 (EC718519) | Syria | 61 | ILWL 203 (EC789113) | Syria |
| 14 | ILWL 246 (EC718521) | Syria | 62 | ILWL 235 (EC718291) | Syria |
| 15 | ILWL 278 (EC718529) | Turkey | 63 | ILWL 320 (EC718295) | Turkey |
| 16 | ILWL 330 (EC718548) | Syria | 64 | ILWL 357 (EC718297) | Syria |
| 17 | ILWL 343 (EC718554) | Syria | 65 | ILWL 361 (EC718298) | Syria |
| 18 | ILWL 344 (EC718555) | Syria | 66 | ILWL 409 (EC718301) | Syria |
| 19 | ILWL 349 (EC718560) | Syria | 67 | ILWL 436 (EC718302) | Turkey |
| 20 | ILWL 359 (EC718566) | Syria | 68 | ILWL 438 (EC718303) | Turkey |
| 21 | ILWL 384 (EC718585) | Tajikistan | 69 | EC718311 | Israel |
| 22 | ILWL 443 (EC718596) | Turkey | 70 | EC718694 | Syria |
| 23 | ILWL 476 (EC718605) | Turkey | | *L. ervoides* | |
| 24 | ILWL 480 (EC718609) | Syria | 71 | EC718692 | Spain |
| | *L. nigricans* | | 72 | ILWL 43 (EC718316) | Croatia |
| 25 | ILWL 9 (EC718236) | Syria | 73 | ILWL 50 (EC718321) | Croatia |
| 26 | ILWL 14 (EC718267) | Syria | 74 | ILWL 51 (EC718322) | Montenegro |
| 27 | ILWL 15 (EC718238) | France | 75 | ILWL 58 (EC718329) | Turkey |
| 28 | ILWL16 (EC718239) | Alpes-Cote d'Azur | 76 | ILWL 60 (EC718331) | Turkey |
| 29 | ILWL 18 (EC718241) | France | 77 | ILWL 61 (EC718332) | Turkey |
| 30 | ILWL 19 (EC718242) | Spain | 78 | ILWL 63 (EC718333) | Turkey |
| 31 | ILWL 22 (EC718245) | Italy | 79 | ILWL 65 (EC718335) | Turkey |
| 32 | ILWL 31 (EC718254) | Spain | 80 | ILWL 66 (EC718336) | Turkey |
| 33 | ILWL 34 (EC718256) | Ukraine | 81 | ILWL 204 (EC718360) | Turkey |
| 34 | ILWL 37 (EC718257) | Turkey | 82 | ILWL 234 (EC718362) | Syria |
| 35 | ILWL 38 (EC718258) | Turkey | 83 | ILWL 269 (EC718370) | Turkey |
| 36 | ILWL 191 (EC718682) | Croatia | 84 | ILWL 276 (EC718374) | Turkey |
| 37 | ILWL 460 (EC718262) | Turkey | 85 | ILWL 292 (EC718378) | Turkey |
| 38 | EC718266 | Italy | 86 | ILWL 321 (EC718383) | Turkey |
| 39 | EC718270 | Croatia | 87 | ILWL 398 (EC718401) | Lebanon |
| 40 | EC718273 | Spain | 88 | ILWL 401 (EC718404) | Lebanon |
| 41 | EC718275 | Turkey | 89 | ILWL 408 (EC718407) | Syria |
| | *L. tomentosus* | | 90 | ILWL 414 (EC718411) | Syria |
| 42 | ILWL 90 (EC718471) | Turkey | 91 | ILWL 418 (EC718413) | Syria |
| 43 | ILWL 195 (EC718682) | Syria | 92 | ILWL 441 (EC718417) | Turkey |
| 44 | ILWL 198 (EC718685) | Syria | 93 | ILWL 442 (EC718418) | Turkey |

*(Continued)*

**Table 1.** (Continued)

| S. No | Taxa/accession | Origin | S. No. | Taxa/accession | Origin |
|---|---|---|---|---|---|
| 45 | ILWL 199 (EC718686) | Syria | 94 | EC718439 | Israel |
| 46 | ILWL 305 (EC728782) | Turkey | | *L. culinaris* | |
| 47 | ILWL 308 (EC728782) | Turkey | 95 | ILL 8006 | Syria |
| 48 | ILWL 480 (EC741251) | Syria | 96 | ILL 10829 | Syria |
| 49 | EC718673 | Syria | | | |

## Phenotypic diversity and association study

The phenotypic diversity analysis was carried out using quantitative data of 96 promising accessions of wild lentil. Euclidean distances were calculated using quantitative data which was used to establish relationship between interspecific accessions. DARwin 5.0 software of Perrier et al. [13] was used for hierarchical clustering of accessions using UPGMA mode based upon Jaccard's coefficient. Correlation among various parameters was determined using SAS software [9].

## Screening against rust, powdery mildew and Fusarium wilt

**Rust (*Uromyces fabae* (Grev.) Fuckel).** The experiment was conducted at CSKHPKV Research and Extension Centre, Dhaulakuan ($30^0$ 49´ N, $77^0$ 59´ E, altitude 468 m amsl) during winter season of 2014–2015, 2015–2016 and 2016–2017. All the test accessions and rust susceptible check 'var. L830' were sown in plastic pots (20 cm diameter) filled with a mixture of top field soil and farm yard manure (10:1) in two replications. The pots with plants at pre-flowering stage (~60 days after sowing) of wild accessions were shifted in the field sown with rust susceptible varieties, K 75 and L 830 from India. The test accessions were frequently inoculated by spraying suspension of local isolate of *U. viciae fabae* ($1 \times 10^6$ spores/ml). The data were recorded using 1–9 rating scales at vegetative and reproductive stages as followed by Mayee and Datar [14].

**Powdery mildew (*Erysiphe trifolii*).** Another experiment was undertaken at CSKHPKV Research and Extension Centre, Dhaulakuan during winter season of 2014–2015, 2015–2016 and 2016–2017. The plants of test accessions were raised in plastic pots in a mixture of soil and farmyard manure mixture (10:1) under green house conditions. The disease symptoms started developing at flower initiation stage and susceptible checks developed disease earlier and faster. The pots with heavily infected susceptible plants were transferred near the test accessions raised in the green house. The heavily infected plants were shaken with the wooden stick to dislodge the conidia which dispersed and infected the healthy plant parts of the test accessions as also suggested by Tiwari et al. [15]. The full infection of susceptible accessions was observed near green pod stage and scorings were done at different intervals using 1–9 rating scales [15].

**Fusarium wilt (*Fusarium oxysporum* f. sp. *lentis* (Vasd. Srin.) Gord).** All 96 wild lentil accessions were screened against Fusarium wilt resistance at Rafi Ahmad Kidwai Agriculture College, Sehore, Madhya Pradesh ($23^0$ 12´ N, $77^0$ 05´ E, altitude 502 m amsl) during winter season of 2014–2015 and 2015–2016. For screening of wild lentil accessions, protocol was standardized as per the method described by Bayaa and Erskine [16–19]. The performance of disease pressure was compared with the resistant (PL639) and susceptible (L-9-12) checks from India, which were repeated after 15 accessions of each replication. The data were taken using 1–9 rating scales as followed by Bayaa et al. [17, 19].

# Results

## Diversity for quantitative characters

The analysis of variance revealed that the wild lentil accessions differed significantly in terms of seed yield plant[-1] and its important component characters (significant at $p = 0.05$) and also manifested by the range, mean and coefficient of variation for all the metric characters assessed. However, the frequency distribution of important quantitative characters depicted in Fig 1 exhibited that a wide range of variation was observed in plant height, number of branches plant[-1], number of pods plant[-1] and seed yield plant[-1]. In plant height, all 96 accessions were grouped into different classes (Fig 1) and only two accessions ILWL15 (63.50 cm) and ILWL19 (66.50 cm) of *L. nigricans* were reported for maximum height. Likewise, number of branches plant[-1] was highest in accession ILWL 418 (52 branches) of *L. ervoides* and ILWL 480 (37 branches) of *L. orientalis*. The number of pods plant[-1] ranged from 3 to 1167 pods plant[-1] and only two accessions ILWL 418 (1002 pods) of *L. ervoides* and ILWL 18 (1167 pods) of *L. nigricans* revealed maximum number. The character seed yield plant[-1] also showed variation and only two accessions ILWL 18 (13.60 g) and ILWL 19 (12.55 g) of *L. nigricans* showed maximum seed yield palnt[-1].

## Diversity analysis

Wild lentil accessions formed two major clusters I and II in hierarchical clustering as shown in Fig 2. Major cluster I grouped only three accessions including ILWL418 (*L. ervoides*) from

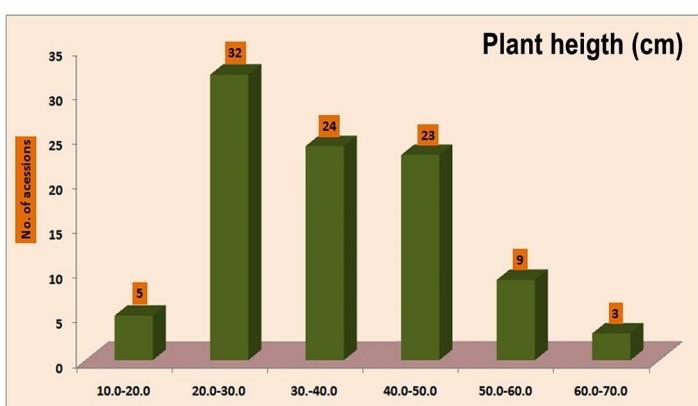
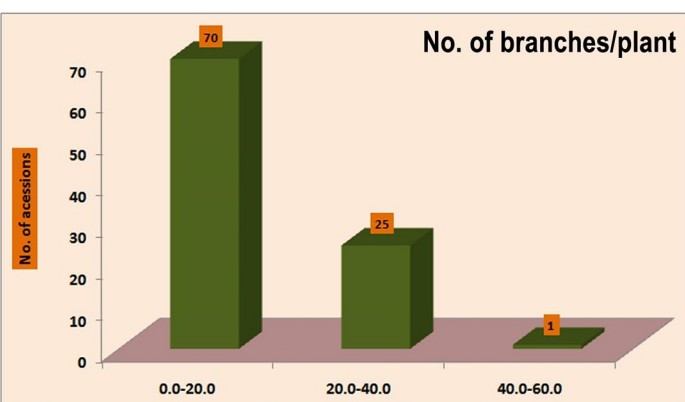
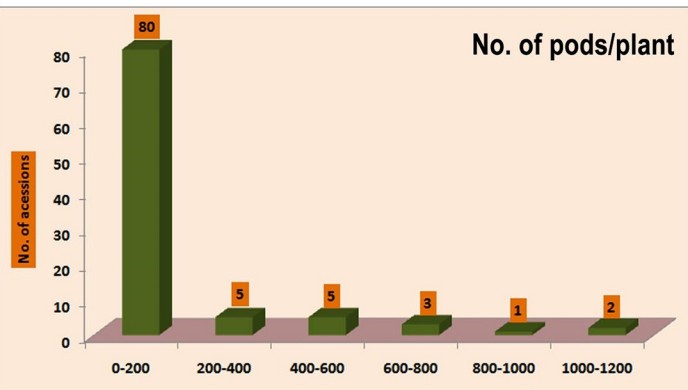
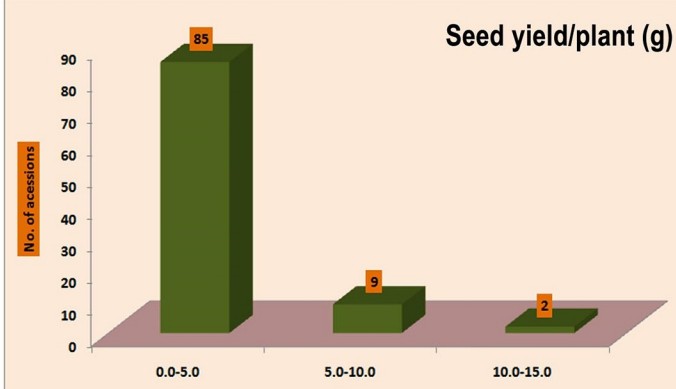

**Fig 1. Diversity of wild lentil accessions for important quantitative traits.**

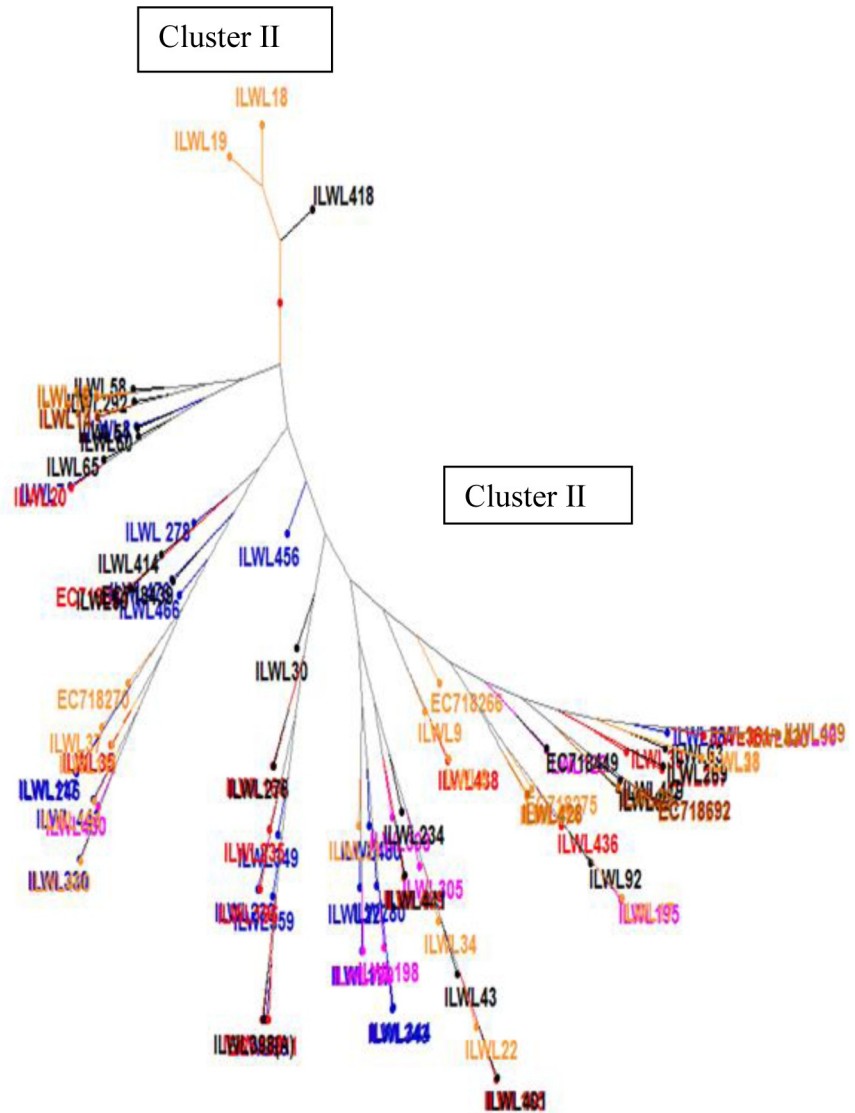

**Fig 2. Hierarchical clustering of wild lentil accessions based on quantitative data analysis.**

Syria; ILWL18 and ILWL19 (*L. nigricans*) from France and Spain, respectively. Cluster II contains all remaining accessions was further divided into A and B clusters. Bifurcation of Cluster A resulted into two sub clusters AI and AII, where sub cluster AI was occupied by *L. ervoides* accessions ILWL58 and ILWL292 from Turkey; *L. nigricans* accessions ILWL16 from Alpes Cote d' Azur and ILWL15 from France; and ILWL14 and ILWL15 of *L. lamottei* from France. Sub cluster AII grouped ILWL7 and ILWL8 of *L. orientalis* from Turkey; ILWL51, ILWL60 and ILWL65 of *L. ervoides* including first from Montenegro and other two from Turkey; and ILWL20 of *L. odemensis* from Palestine. Cluster B was occupied by two sub clusters BI and BII, each of which were further divided into two groups a, b and c, d, respectively. Group d further formed outgroups (d1 and d2) and finally clutches (d2a and d2b). As Cluster II grouped sub-clusters were occupied by accessions from different wild lentil species belonging to different countries of origin, they showed highest diversity (S1 Table). Euclidean dissimilarity matrices varied from 6.01 between ILWL429 (*L. lamottei*) from Spain and ILWL90 (*L. tomentosus*)

**Table 2. Correlations among various agro-morphological traits of wild lentil accessions.**

| Traits | DF | DM | PH | NBPP | NPPP | NSPPL | NSPPD | SDWT | SYPP | BYPP | HI | RT | PM | FW |
|---|---|---|---|---|---|---|---|---|---|---|---|---|---|---|
| **DF** | 1.00 | .750(**) | -0.13 | -.219(*) | -.542(**) | -.534(**) | -0.16 | -0.06 | -.495(**) | -.311(**) | -.543(**) | -0.15 | -0.06 | -0.15 |
| **DM** | .750(**) | 1.00 | 0.04 | -0.15 | -.346(**) | -.319(**) | -.256(*) | -0.09 | -.350(**) | -.235(*) | -.319(**) | -0.02 | 0.03 | -0.17 |
| **PH** | -0.13 | 0.04 | 1.00 | .242(*) | .360(**) | .411(**) | -0.11 | -0.15 | .468(**) | .344(**) | .438(**) | .342(**) | 0.15 | -0.14 |
| **NBPP** | -.219(*) | -0.15 | .242(*) | 1.00 | .544(**) | .536(**) | -.285(**) | -0.09 | .504(**) | .539(**) | .388(**) | .328(**) | .272(**) | 0.08 |
| **NPPP** | -.542(**) | -.346(**) | .360(**) | .544(**) | 1.00 | .987(**) | -0.15 | -0.09 | .928(**) | .833(**) | .738(**) | .367(**) | .290(**) | 0.03 |
| **NSPPL** | -.534(**) | -.319(**) | .411(**) | .536(**) | .987(**) | 1.00 | -0.18 | -0.08 | .956(**) | .820(**) | .788(**) | .308(**) | .322(**) | -0.02 |
| **NSPPD** | -0.16 | -.256(*) | -0.11 | -.285(**) | -0.15 | -0.18 | 1.00 | -0.12 | -.244(*) | -.247(*) | -0.16 | -.291(**) | -.224(*) | 0.13 |
| **SDWT** | -0.06 | -0.09 | -0.15 | -0.09 | -0.09 | -0.08 | -0.12 | 1.00 | 0.00 | -0.12 | 0.02 | -0.02 | 0.07 | 0.14 |
| **SYPP** | -.495(**) | -.350(**) | .468(**) | .504(**) | .928(**) | .956(**) | -.244(*) | 0.00 | 1.00 | .814(**) | .841(**) | .373(**) | .391(**) | 0.00 |
| **BYPP** | -.311(**) | -.235(*) | .344(**) | .539(**) | .833(**) | .820(**) | -.247(*) | -0.12 | .814(**) | 1.00 | .508(**) | .377(**) | .283(**) | -0.02 |
| **HI** | -.543(**) | -.319(**) | .438(**) | .388(**) | .738(**) | .788(**) | -0.16 | 0.02 | .841(**) | .508(**) | 1.00 | .312(**) | .347(**) | 0.00 |
| **Rust** | -0.15 | -0.02 | .342(**) | .328(**) | .367(**) | .308(**) | -.291(**) | -0.02 | .373(**) | .377(**) | .312(**) | 1.00 | .314(**) | -0.08 |
| **PM** | -0.06 | 0.03 | 0.15 | .272(**) | .290(**) | .322(**) | -.224(*) | 0.07 | .391(**) | .283(**) | .347(**) | .314(**) | 1.00 | -0.02 |
| **FW** | -0.15 | -0.17 | -0.14 | 0.08 | 0.03 | -0.02 | 0.13 | 0.14 | 0.00 | -0.02 | 0.00 | -0.08 | -0.02 | 1.00 |

*Correlation is significant at the 0.05 level

**Correlation is significant at the 0.01 level

DF, Days to flowering; DM, Days to maturity; PH, Plant height; NBPP, Number of branches plant[-1]; NPPP, Number of pods plant[-1]; NSPPL, Number of seeds plant[-1]; NSPPD, Number of seeds pod[-1]; SDWT, Seed weight; SYPP, Seed yield plant; BYPP, Biological yield plant; HI, Harvest index; RT, Rust; PM, Powdery mildew; FW, Fusarium wilt

from Turkey to 2156.52 between ILWL18 (*L. nigricans*) from France and ILWL476 (*L. orientalis*) from Turkey (S1 Table). First two principal coordinates in factorial analysis explained 99.07% and 0.52% variability, respectively. In both type of groupings, the role of geographical or country of origin of accessions was not observed.

## Correlations among different parameters

The correlation coefficient among different agro-morphological traits of wild lentil accessions is shown in Table 2. Correlation indices highlighted a number of significant inter-relationships (both +ve and -ve) among studied traits. In the present study, significant positive correlation was observed between days to flowering and maturity (>0.4 r-value). Similarly, significant positive correlations were obtained between seed yield plant[-1], biological yield plant[-1] and harvest index. These traits also showed positive correlations with plant height ($r^2$ <4.0 in case of biological yield plant[-1]), number of branches plant[-1] ($r^2$ <4.0 in case of harvest index), number of pods plant[-1] and seeds plant[-1]. Seeds plant[-1] showed significant correlation with plant height, branches plant[-1] and pods plant[-1]. The number of pods plant[-1] exhibited significant positive correlation with branches plant[-1]. Significant negative correlation was observed between days to flowering and seed yield plant[-1], seeds plant[-1], pods plant[-1], and harvest index.

## Principle component analysis

Maximum variability (66.31%) was explained by only three components with Eigenvalue >1.0 (Table 3). Of the total variation, 42.37% was exhibited by PC1, and the contributing traits with high coefficients included seed yield plant[-1], seeds plant[-1], pods plant[-1], harvest index and biological yield plant[-1]. Similarly, 15.04% variability was explained by PC2, and the contributing traits with high coefficients included days to maturity, days to flowering, rust incidence and plant height. Similarly, PC3 contributed 8.89% of overall variability, and the traits contributing

**Table 3. Eigen vectors, eigenvalues, individual and cumulative percentages of variation explained by the first three principal components (PC) of wild lentil accessions.**

| Traits | Prin1 | Prin2 | Prin3 |
|---|---|---|---|
| Days to flowering | -0.25 | 0.49 | 0.00 |
| Days to maturity | -0.19 | 0.56 | -0.01 |
| Plant height (cm) | 0.20 | 0.22 | -0.30 |
| Number of branches plant$^{-1}$ | 0.25 | 0.12 | 0.09 |
| Number of pods plant$^{-1}$ | 0.39 | -0.05 | -0.06 |
| Number of seeds plant$^{-1}$ | 0.39 | -0.02 | -0.06 |
| Number of seeds pod$^{-1}$ | -0.10 | -0.41 | -0.37 |
| 100 -seed weight (g) | -0.02 | -0.09 | 0.71 |
| Seed yield plant$^{-1}$ (g) | 0.40 | 0.04 | 0.02 |
| Biological yield plant$^{-1}$ (g) | 0.34 | 0.10 | -0.06 |
| Harvest index (%) | 0.35 | -0.04 | 0.02 |
| Rust severity | 0.26 | 0.26 | 0.04 |
| Powdery mildew severity | 0.17 | 0.22 | 0.33 |
| Fusarium wilt | 0.00 | -0.26 | 0.37 |
| Eigen value | 5.93 | 2.11 | 1.24 |
| Percent | 42.37 | 15.04 | 8.89 |
| Cum. Percentage | 42.37 | 57.42 | 66.31 |

to diversity included 100- seed weight, Fusarium wilt and Powdery mildew severity. Fig 3A and 3B shows the two dimensional variability pattern and contributing traits towards variability, respectively.

## Resistance to key biotic stresses

**Rust resistance.** Majority of wild lentil accessions were resistant and moderately resistant against rust under controlled artificial inoculation conditions. Accessions ILWL90, ILWL195, ILWL198 and ILWL480 (*L. tomentosus*); ILWL230, 349, 476 (*L. orientalis*); ILWL203, 235 and 357 (*L. odemensis*); ILWL60, 204, 292, 414 and 442 (*L. ervoides*); and ILWL16, 18, 37, 460 and EC718266 (*L. nigricans*) were identified as resistant (Tables 4 and 5). The remaining accessions

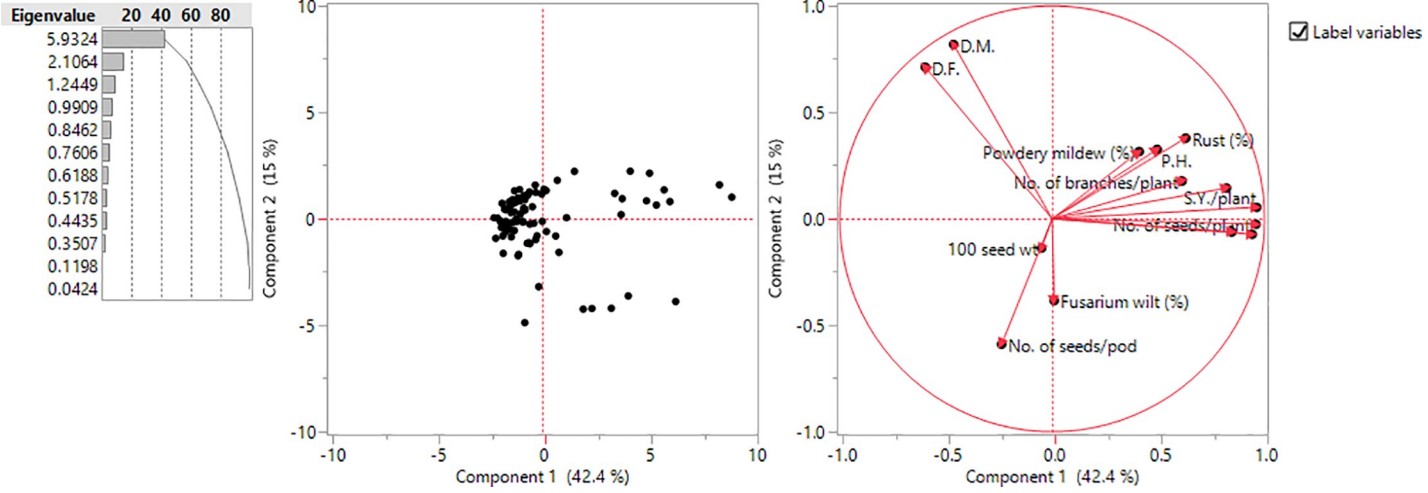

**Fig 3.** PCA analysis—(a) 2-D graph of first two principal components and (b) various traits contributing to the variability.

**Table 4. Range of variation of wild lentil accessions against rust, powdery mildew and *Fusarium* wilt.**

| Species | Rust | | | Powdery mildew | | | Fusarium wilt | | |
|---|---|---|---|---|---|---|---|---|---|
| | Range | Mean ±SE | CV (%) | Range | Mean ±SE | CV (%) | Range | Mean ±SE | CV (%) |
| *L. culinaris* | 1–1 | 1.00±0.00 | 0.00 | 0–5 | 5.00±0.00 | 0.00 | 1–7 | 4.00±3.00 | 106.07 |
| *L. orientalis* | 1–9 | 3.41±0.70 | 84.23 | 0–5 | 3.56±0.35 | 42.29 | 1–9 | 4.44±0.74 | 70.40 |
| *L. tomentosus* | 1–5 | 1.86±0.59 | 84.73 | 0–5 | 2.29±0.75 | 86.45 | 1–9 | 5.00±1.15 | 61.10 |
| *L. odemensis* | 1–6 | 2.33±0.40 | 66.13 | 0–4 | 1.60±0.39 | 93.90 | 1–9 | 5.67±0.64 | 43.57 |
| *L. ervoides* | 1–8 | 1.95±0.42 | 92.93 | 0–5 | 1.90±0.45 | 105.12 | 1–9 | 5.18±0.72 | 65.14 |
| *L. nigricans* | 1–7 | 2.10±0.46 | 98.81 | 0–5 | 2.52±0.52 | 95.11 | 1–9 | 4.43±0.69 | 71.61 |
| *L. lamottei* | 1–7 | 3.43±0.97 | 75.04 | 0–5 | 3.29±0.75 | 60.14 | 1–9 | 4.43±1.04 | 62.33 |

were scored as moderately resistant, susceptible and some of them were highly susceptible against the pathogen. The distribution of accessions into various reaction categories based on disease score is presented in Fig 4.

**Powdery mildew resistance.** Out of 24 accessions of *L. orientalis*, ILWL 230 and ILWL 476 were found highly resistant. Likewise, in *L. odemensis*, accessions ILWL 39, ILWL 203, and IG 136788; ILWL 198 and ILWL 480 of *L. tomentosus* were also found to be highly resistant (Tables 4 and 5). However, accessions ILWL 51, ILWL 401, ILWL 418 and ILWL 441 of *L. ervoides*; ILWL9, ILWL22, ILWL34, ILWL37 and ILWL191 of *L. nigricans*; and ILWL 29 of *L.*

**Table 5. Identification of donor accessions for their introgression to lentil genetic improvement against important agronomic and major biotic stress related traits.**

| Taxa/ trait | Accession | Gene pool | Country origin |
|---|---|---|---|
| **Agronomic** | | | |
| *L. orientalis* | ILWL255, 476 ((High pods plant$^{-1}$, shorter internode) | Primary | Turkey, Syria |
| *L. odemensis* | ILWL203, 237 (High pods plant$^{-1}$, shorter internode) | Primary | Syria |
| *L. ervoides* | ILWL294, 321 ((High pods plant$^{-1}$, shorter internode) | Tertiary | Turkey, Syria |
| **Rust** | | | |
| *L. orientalis* | ILWL230, 349, 466 | Primary | Turkey |
| *L. odemensis* | ILWL235, 357, ILWL203 | Primary | Syria |
| *L. tomentosus* | ILWL 90, 95,198, 308, 480 | Primary | Syria |
| *L. ervoides* | ILWL60, 92, 269, 292, 414, 419, IG136226 | Tertiary | Turkey, Syria |
| *L. nigricans* | ILWL9, 17, 37, 111, 460, IG136639, IG136648 | Quaternary | |
| **Powdery mildew** | | | |
| *L. orientalis* | ILWL 230, 466 | Primary | Syria |
| *L. odemensis* | ILWL 39, 196, IG136788 | Primary | Turkey, Syria |
| *L. tomentosus* | ILWL 198, 480 | Primary | Turkey, Syria |
| *L. ervoides* | ILWL51, 401, 418, 441 | Tertiary | Montenegro, Lebanon, Syria, Turkey |
| *L. nigricans* | ILWL6, 9, 22, 34, 37, 111 | Quaternary | Turkey, Syria Italy, Ukraine, Turkey, Turkey |
| *L. lamottei* | ILWL 29 | Secondary | Spain |
| **Fusarium wilt** | | | |
| *L. orientalis* | ILWL7, 96, 117, 227, 246, 344, 359 | Primary | Turkey, Turkey, Syria, Syria, Syria, Syria, Syria, |
| *L. odemensis* | ILWL15, 39 | Primary | France, Turkey |
| *L. tomentosus* | ILWL 199, 308 | Primary | |
| *L. ervoides* | ILWL 58, 398, 414, IG136626 | Tertiary | Turkey, Lebanon, Syria Israeli |
| *L. nigricans* | ILWL17, 19, 23, 34, 38, 474 | Quaternary | Spain, Italy, Ukraine, Turkey, Turkey |
| *L. lamottei* | ILWL 20, 430 | Secondary | PAL, Spain, |

Gene pool classification of Wong et al. [31]

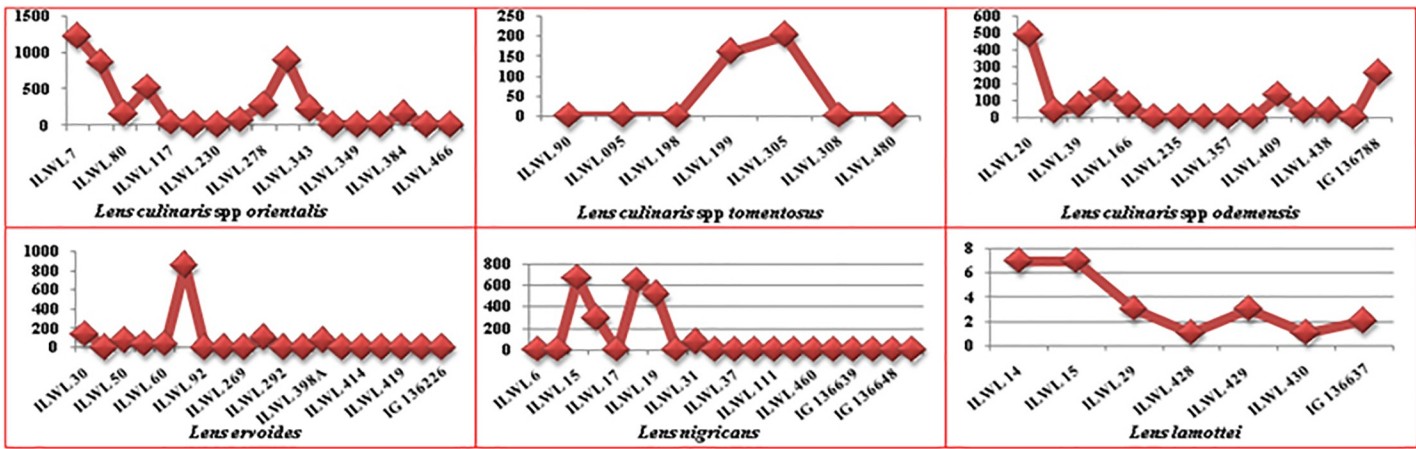

**Fig 4. Screening of wild lentil accessions against rust resistance.**

*lamottei* with disease reaction 2 were found to be resistant. The remaining accessions were either moderately resistant or susceptible against the pathogen (Fig 5).

**Fusarium wilt resistance.** The wild lentil accessions, which showed resistant disease reaction to Fusarium wilt included ILWL7, ILWL96, ILWL117, ILWL227, ILWL246, ILWL344 and ILWL359 (*L. orientalis*); ILWL15 and ILWL39 (*L. odemensis*); ILWL199 and ILWL 308 (*L. tomentosus*). Accessions ILWL58, ILWL398, ILWL414, IG136626 (*L. ervoides*); ILWL17, ILWL19, ILWL23, ILWL34, ILWL38 and ILWL474 (*L. nigricans*) and ILWL20 and ILWL430 (*L. lamottei*) were found resistant against the wilt (Tables 4 and 5). Some accessions were moderately resistant and the remaining was susceptible against the pathogen as depicted in Fig 6.

## Discussion

The precise evaluation of crop wild relatives is pre-requisite to identify target traits of interest followed by their introgression into the background of cultivated varieties for enhancing genetic gains [20–22]. A wide range of variation in wild lentil accessions was observed against the target traits viz. earliness, high pod number and seed yield suggesting diverse genetic makeup and geographical origins of wild lentil collections. These promising accessions belonging to different taxa and ecological niches could be useful resource for enhancing genetic gains

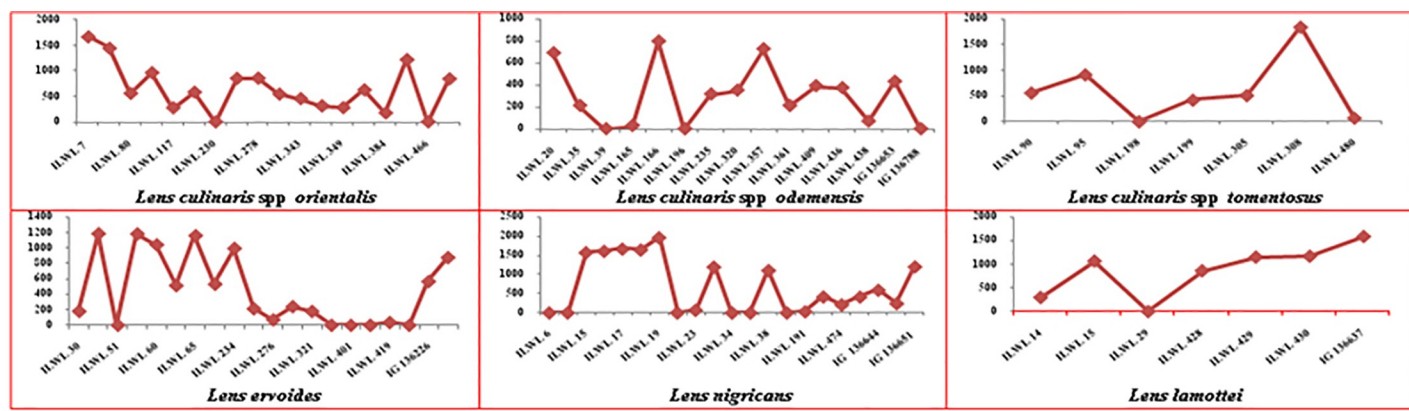

**Fig 5. Screening of wild lentil accessions against powdery mildew resistance.**

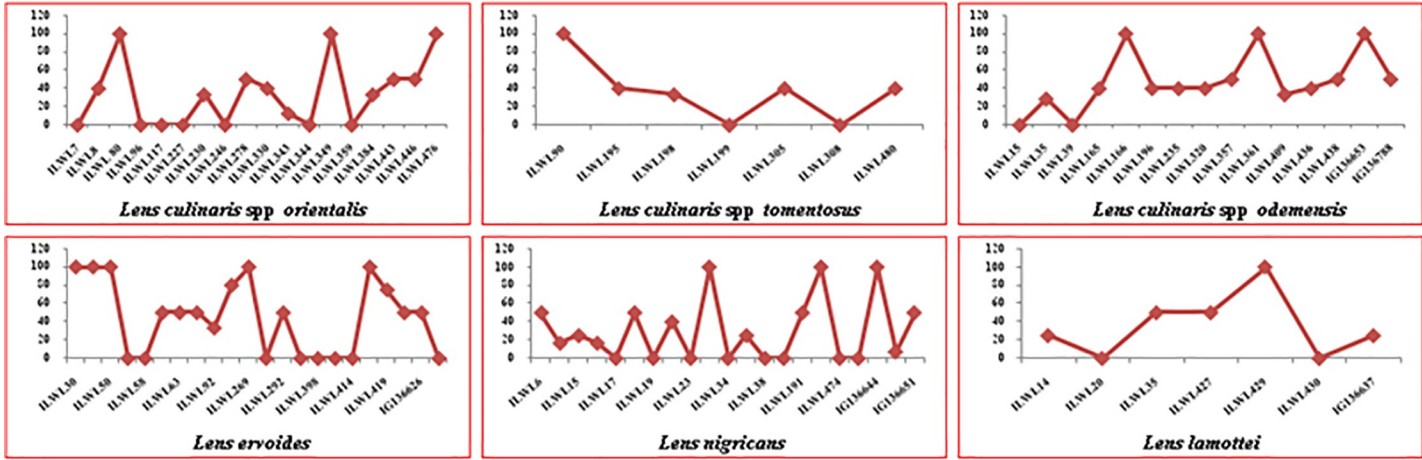

**Fig 6. Screening of wild lentil accessions against Fusarium wilt resistance.**

of cultivated varieties. Accessions ILWL 18 and ILWL 19 of *L. nigricans* were found promising for high seed yield plant[-1], suggesting their broader genetic base as also reflected in clustering pattern of wild accessions based on quantitative data analysis. Euclidean dissimilarity matrices of wild lentil accessions ranged from 6.01 to 2156.52, which revealed very high variability among interspecific lentil taxa. The observed variability could be due to the presence of diverse origins of these wild accessions. Among the different countries of origin, accessions from Turkey showed maximum variability for the studied traits suggesting more exploration and collection of lentil germplasm. Viera et al. [23] characterized wheat germplasm based on phenotypic traits and obtained distances up to 196.61. The report of existing variability among germplasm accessions provides an idea about the expected heterosis for starting breeding programme. A broad range of Euclidean distances obtained in the present study shows the importance of wild lentil accessions as a source of diverse heterotic material [24]. However, the diversity pattern further suggested that no role was played by the geographical distributions of global accessions in their grouping. This grouping pattern can be described on the basis of different genetic constitution of accessions. Further, it might be due to continuous gene flow occurrence among wild lentil taxa [8]. No role of geographical distributions was observed in grouping pattern of accessions clustered on the basis of morphological, biochemical and molecular markers in guar [25]. Similar observation in wild lentil accessions was also reported by Singh et al. [8] and Kumar et al. [26]. Further, significant positive correlation of seed yield plant[-1] with seeds plant[-1], pods plant[-1], branches plant[-1] was observed in lentil accessions and similar correlations have also been reported by other workers [27–29]. They reported that seeds plant[-1], biological yield plant[-1], pods plant[-1], and harvest index are the important yield component traits in lentil contributing towards yield enhancement. Such correlations are useful for lentil genetic improvement and breeding programmes. These observations were found to be in accordance with the results of Toklu et al. [29] and Abo-Hegazy et al. [30]. The traits which contributed heavily towards variability included seed yield plant[-1], seeds pod[-1], pods plant[-1], biological yield plant[-1] and harvest index. These findings highlighted the significance of yield related component characters generating substantial variations in wild lentil accessions.

Furthermore, wild taxa of lentil are known to be resistant against major biotic and abiotic stresses [8]. They are the carrier of important characters and have potential value in diversification of cultivated gene pool for enhancing genetic gains. Some wild lentil accessions revealed resistant disease reaction against more than one disease (Table 5) viz. accessions ILWL230 and

ILWL476 of *L. orientalis* (rust and powdery mildew); accessions ILWL9 and ILWL37 of *L. nigricans* (rust and powdery mildew); accession IG136639 of *L. ervoides* (powdery mildew and Fusarium wilt) and accession ILWL308 of *L. tomentosus* (rust and Fusarium wilt). It is important to mention that the northwestern part has been the traditional lentil cultivation area in India. But in late 1980s, among various factors including the epidemics of rust, powdery mildew and Fusarium wilt resulted in drastic reduction of lentil area and production as well. In this context, gene sources identified against rust, powdery mildew and *Fusarium* wilt in the present study can prove to be useful donors which could be exploited through pre-breeding with the aim to develop usable germplasm for developing resistant lentil varieties, especially for the epidemiologically important Indian conditions. Furthermore, the development of lentil cultivars with combined resistance against major biotic stresses may help in the development of appropriate management strategies in future in conjunction with allelism studies to identify and pyramid novel genes for resistance breeding of lentil.

## Conclusions

During the past few years, little efforts were made on evaluation of wild lentil genetic resources in India. However, this is the first attempt on characterization and evaluation of all available global wild lentil germplasm against the target traits of interest under diverse agro-ecological conditions. Further, the study has helped in identifying the confirmed and stable gene sources (donors) across intra and interspecific accessions viz. ILWL203 (EC789113) of *L. odemensis* for rust and high pod number; ILWL230 (EC718515), ILWL476 (EC718605) of *L. orientalis* for rust and powdery mildew; ILWL191 (EC718682), ILWL9 (EC718236), and ILWL37 (EC718257) of *L. nigricans* for rust and powdery mildew, IG136639 of *L. ervoides* for powdery mildew and Fusarium wilt and ILWL308 (EC728782) of *L. tomentosus* for rust and Fusarium wilt. These gene sources could be taken in lentil wide hybridization programme for enhancing genetic gains of cultivated varieties and could also be shared among lentil breeders/researchers in the country or elsewhere under Standard Material Transfer Agreement (SMTA) for strengthening the on-going lentil genetic improvement programmes.

## Supporting information

**S1 Table. Clustering pattern of wild lentil core accessions based on agro-morphological characters.**
(DOCX)

## Author Contributions

**Conceptualization:** Mohar Singh.

**Data curation:** Sandeep Kumar, Daisy Basandrai, Dorin Gupta.

**Formal analysis:** Sandeep Kumar, Nikhil Malhotra, Dorin Gupta.

**Funding acquisition:** Ashutosh Sarker.

**Methodology:** Ashwani Kumar Basandrai, Nikhil Malhotra, Deep Rattan Saxena, Ashutosh Sarker, Kuldeep Singh.

**Validation:** Ashwani Kumar Basandrai, Daisy Basandrai, Deep Rattan Saxena.

**Writing – original draft:** Mohar Singh.

**Writing – review & editing:** Kuldeep Singh.

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
