## [Decision Letter · Decision Letter 0]

30 Jan 2020

PONE-D-19-35002

Evaluation and identification of wild lentil accessions for enhancing genetic gains of cultivated varieties

PLOS ONE

Dear Dr. Singh,

Thank you for submitting your manuscript to PLOS ONE. After careful consideration, we feel that it has merit but does not fully meet PLOS ONE’s publication criteria as it currently stands. Therefore, we invite you to submit a revised version of the manuscript that addresses the points raised during the review process.

We would request you review the comments and suggestions of the reviewers in the attached manuscripts and make necessary corrections requested by them.

We would appreciate receiving your revised manuscript by Mar 15 2020 11:59PM. To enhance the reproducibility of your results, we recommend that if applicable you deposit your laboratory protocols in protocols.io, where a protocol can be assigned its own identifier (DOI) such that it can be cited independently in the future. For instructions see: http://journals.plos.org/plosone/s/submission-guidelines#loc-laboratory-protocols

We look forward to receiving your revised manuscript.

Kind regards,

Prasanta K. Subudhi, Ph.D.

Academic Editor

PLOS ONE

Journal Requirements:

The authors also thank the Department of Agriculture Cooperation and Farmer’s Welfare, Govt. of India for grant in aid to accomplish the proposed objectives.

AS; Ashutosh Sarker, ICARDA office will make publication charges available with him

Additional Editor Comments:

Minor revision

Reviewers' comments:

Reviewer's Responses to Questions

**Comments to the Author**

1. Is the manuscript technically sound, and do the data support the conclusions?

Reviewer #1: Yes

Reviewer #2: Partly

2. Has the statistical analysis been performed appropriately and rigorously? 

Reviewer #1: Yes

Reviewer #2: I Don't Know

3. Have the authors made all data underlying the findings in their manuscript fully available?

Reviewer #1: Yes

Reviewer #2: Yes

4. Is the manuscript presented in an intelligible fashion and written in standard English?

Reviewer #1: Yes

Reviewer #2: No

5. Review Comments to the Author

Reviewer #1: The work was implemented properly; data analyses is okay. The work also contributes in identifying wild lentil accession that can be exploited to develop high yielding genotypes with multiple resistance to key diseases. Hence worth to publish

Reviewer #2: In plant materials, you mentionned check varieties : Precoz and L830. Kindly add their origin in table 1 and why did you choose those 2 varieties.

Did K75 was used as check variety for Rust ?

Did PL639 and L-9-12 were used as check varieties for Fusarium wilt ?

Kindly add their origin

6. PLOS authors have the option to publish the peer review history of their article (what does this mean?). If published, this will include your full peer review and any attached files.

Reviewer #1: Yes: seid ahmed kemal

Reviewer #2: No

---

## [Author Response · Author response to Decision Letter 0]

5 Feb 2020

Dear Academic Editor and Reviewers

We tried our level best to improve the proposed research article on wild lentil and necessary changes were made in the revised text using track change mode.

Reviewer 1

Although as such no comments were mentioned separately, we have included all changes made in the manuscript as suggested and we really appreciate the changes suggested in the paper. 

Reviewer 2

Comment: In plant materials, you mentioned check varieties: Precoz and L830. Kindly add their origin in table 1 and why did you choose those 2 varieties.

Reply: The country of origin for these two varieties has been mentioned in revised text in the Material and method section. As far as taking these varieties as checks are concerned, these lentil varieties are having very good agronomic base in majority of lentil growing areas in India.

Comment: Did K75 was used as check variety for Rust? 

Reply: Yes for rust susceptibility 

Comment: Did PL639 and L-9-12 were used as check varieties for Fusarium wilt? Kindly add their origins

Reply: Yes taken as checks and origins have been added in the modified text.

---

## [Editor Report · Decision Letter 1]

11 Feb 2020

Evaluation and identification of wild lentil accessions for enhancing genetic gains of cultivated varieties

PONE-D-19-35002R1

Dear Dr. Singh,

We are pleased to inform you that your manuscript has been judged scientifically suitable for publication and will be formally accepted for publication once it complies with all outstanding technical requirements.

With kind regards,

Prasanta K. Subudhi, Ph.D.

Academic Editor

PLOS ONE

Additional Editor Comments (optional):

Accept
---

## [Editor Report · Acceptance letter]

13 Feb 2020

PONE-D-19-35002R1 

Evaluation and identification of wild lentil accessions for enhancing genetic gains of cultivated varieties 

Dear Dr. Singh:

I am pleased to inform you that your manuscript has been deemed suitable for publication in PLOS ONE. Congratulations! Your manuscript is now with our production department. 

With kind regards,

on behalf of

Dr. Prasanta K. Subudhi 

Academic Editor

PLOS ONE